# Bone Marrow Mastocytosis: A Diagnostic Challenge

**DOI:** 10.3390/jcm10071420

**Published:** 2021-04-01

**Authors:** Roberta Zanotti, Ilaria Tanasi, Andrea Bernardelli, Giovanni Orsolini, Patrizia Bonadonna

**Affiliations:** 1Hematology Unit, Department of Medicine, Azienda Ospedaliera Universitaria Integrata di Verona, 37134 Verona, Italy; ilaria_tanasi@hotmail.com (I.T.); bernardelli89@gmail.com (A.B.); 2Gruppo Interdisciplinare per lo Studio della Mastocitosi (GISM), Azienda Ospedaliera Universitaria Integrata di Verona, 37134 Verona, Italy; giovanni.orsolini@aovr.veneto.it (G.O.); patrizia.bonadonna@aovr.veneto.it (P.B.); 3Reumathology Unit, Department of Medicine, Azienda Ospedaliera Universitaria Integrata di Verona, 37134 Verona, Italy; 4Allergy Unit, Department of Medicine, Azienda Ospedaliera Universitaria Integrata di Verona, 37134 Verona, Italy

**Keywords:** bone marrow mastocytosis, anaphylaxis, Hymenoptera venom allergy, osteoporosis

## Abstract

Bone marrow mastocytosis (BMM) represents a provisional, indolent subvariant of systemic mastocytosis (SM). Utilizing WHO criteria, BMM requires bone marrow (BM) involvement and the absence of mastocytosis skin lesions. BMM is characterized by male sex prevalence, a slight increase of serum tryptase levels, low BM mast cells (MC) burden, and an indolent clinical course. BMM shows a strong correlation with severe anaphylaxis, mainly due to an IgE-mediated allergy to bee or wasp venom and, less frequently, to unexplained (idiopathic) anaphylaxis. Furthermore, BMM is often associated with osteoporosis which could be the only presenting symptom of the disease. BMM is an undervalued disease as serum tryptase levels are not routinely measured in the presence of unexplained osteoporosis or anaphylaxis. Moreover, BMM patients are often symptom-free except for severe allergic reactions. These factors, along with typical low BM MCs infiltration, may contribute to physicians overlooking BMM diagnosis, especially in medical centers that lack appropriately sensitive diagnostic techniques. This review highlights the need for a correct diagnostic pathway to diagnose BMM in patients with suspected symptoms but lacking typical skin lesions, even in the case of normal serum tryptase levels. Early diagnosis may prevent potential life-threatening anaphylaxis or severe skeletal complications.

## 1. Introduction

Mastocytosis consists of clonal disorders characterized by an abnormal proliferation of mast cells (MC) infiltrating various tissues, particularly skin, bone marrow (BM), and other extracutaneous organs. Depending on the type of affected tissue, mastocytosis can be classified into cutaneous mastocytosis (CM), skin-limited, and systemic mastocytosis (SM) that ranges from an indolent to a very aggressive course [1]. Whereas most patients with CM are children, the majority of adult patients with mastocytosis have SM [1]. In recent years, knowledge of the pathophysiology of mastocytosis has improved. Particularly, research has found that a recurrent activating D816V *KIT* mutation, located in the phosphotransferase domain (PTD) of the receptor, is detectable in more than 80% of adult patients with SM [2,3]. Although rarely seen, SM patients may present with *KIT* mutations other than D816V (i.e., D816H, D816Y) or even mutations outside the PTD region of the *KIT* receptor [4]. In advanced forms of SM, additional genetic alterations involving genes implicated in the pathogenesis of myeloid malignancies, such as mutations of *SRSF2*, *ASXL1*, *RUNX1* (gathered by the S/A/R acronym), *EZH*2, or *CBL*, have been described. These mutations are associated with poor prognosis and more aggressive disease [5,6,7].

For many years, mastocytosis has been considered an orphan disease. Few studies have explored the incidence and prevalence of SM. A Danish group reported an SM prevalence of 9.56 per 100,000 persons, whereas Van Doormaal et al. reported a prevalence of ISM of 13 per 100,000 inhabitants [8,9]. We now know that these data are probably underestimated due to a recent, improved knowledge of the disease, as well as more widely available diagnostic tests (i.e., sensitive *KIT* mutation analysis and flow cytometry techniques) that allow for more precise diagnoses [2,3,10,11,12,13,14,15,16]. Based on our Multidisciplinary Outpatients Clinic for Mastocytosis database, we documented the prevalence of SM as 21 per 100,000 inhabitants in the Verona province adult population (unpublished data).

The diagnosis of mastocytosis relies on the identification of atypical MC in the affected tissue, according to the well-established World Health Organization (WHO) criteria. These criteria are based on morphological, histological, cytofluorimetric, and molecular features [17,18]. The 2016 WHO classification recognizes different clinical variants of mastocytosis: CM, indolent SM (ISM), smoldering SM (SSM), SM with associated hematological neoplasm (SM-AHN), aggressive SM (ASM), mast cell leukemia (MCL), and mast cell sarcoma [17]. CM, ISM, and SMM are considered nonadvanced forms of mastocytosis, while ASM, SM-AHN, and MCL are included in the advanced SM group.

The diagnosis of SM requires the presence of the major histological criterion, which consists of at least 15 multifocal dense MCs infiltrates in the BM or other extracutaneous organs, in addition to at least one minor criterion: (i) presence of atypical morphology in more than 25% of BM or extracutaneous MCs; (ii) serum tryptase (sT) >20 ng/mL; (iii) CD2 and/or CD25 positive MCs in BM or other extracutaneous organs; (iv) detection of *KIT* mutation at codon 816 in BM, blood or extracutaneous organs. Without the major criterion, a diagnosis of SM requires at least three out of four minor criteria [17,18]. SM patients can be further subclassified depending on the presence of B or C findings, defining MC burden or disease aggressiveness, respectively (see Table 1) [17,18].

## 2. Bone Marrow Mastocytosis

The discovery of MC occurred in the 1870s, and was followed by the first report of urticaria pigmentosa (UP). Since then, mastocytosis has been historically considered a skin-limited disease. While SM was first recognized in the mid-20th century, the absence of skin involvement is associated with aggressive behavior because it is typically absent in about 50% of ASM and 30% of SM-AHN and MCL [19,20].

In 1991, a variant of ISM without skin lesions was described [21]. In 2008, the WHO Classification introduced an indolent subvariant of SM without skin involvement, provisionally named bone marrow mastocytosis (BMM) [22]. In 2010, Pardanani et al. showed that BMM represented nearly one-third of all ISM cases [23]. Later, BMM patients’ series have been described by our group and reported in several recent studies [13,14,15,24,25,26,27,28]. However, the prevalence of this entity varies from 10–49% among different series of adult mastocytosis patients. This finding suggests that BMM is a frequently disregarded disease in clinical practice [19,28,29].

The diagnosis of BMM is based on the following criteria of SM: the involvement of the BM, absence of skin lesions, less than two B-finding, absence of any C-finding, and less than 20% of MC at BM smear (Table 1) [17,18]. Since skin lesions are often absent in patients with ASM, SM-AHN, and MCL, it is of particular importance to distinguish between BMM and advanced SM, which generally have constitutional symptoms, altered blood counts, or/and organomegaly [17,18].

BMM is more common in males with normal or slightly elevated sT levels and fewer MC-related symptoms than other ISM types [14,25]. Progression-free survival of BMM is generally good [23], which is consistent with the restriction to the MC compartment of the *KIT* mutation demonstrated in the vast majority of patients with BMM and allergic symptoms [2,25,30,31]. Despite the indolent course, BMM is strongly associated with anaphylactic reactions and potentially life-threatening events. Thus, the early recognition of BMM is advisable.

The diagnosis of BMM may be challenging for clinicians. First, in the absence of skin lesions, SM is suspected based on extremely heterogeneous symptoms. Presenting symptoms of the disease may range from skin mediator-related symptoms (i.e., pruritus, facial flushing, dizziness), gastrointestinal symptoms (i.e., nausea, diarrhea, vomiting, abdominal pain), musculoskeletal and neurological symptoms (i.e., myalgia, headache, fractures, bone pain), constitutional symptoms (i.e., sweating, weight loss, fever), hypotension, and life-threatening anaphylaxis. Several factors may induce the release of MC mediators, such as physical factors, temperature variations, emotional stress, food, alcohol consumption, certain drugs, and Hymenoptera bites. Furthermore, the suspicion of SM is frequently derived from unexplained osteoporosis with or without fragility fractures, especially among males [32].

Diagnosing BMM is also difficult due to the high percentage of BMM patients who do not satisfy the major histological criterion but, instead, present with isolated, atypical, or small aggregates of BM MCs. In these cases, the diagnosis relies both on the pathologist’s experience and on a correct clinical suspicion. Additionally, sT levels in BMM patients could be normal or only slightly elevated, which is consistent with a very low MC burden in those with BM [13,15]. Consequently, the diagnosis of BMM is often based on at least three minor WHO criteria. In some cases with very low MC burden, sufficient criteria for diagnosis of SM are not reached, even in the presence of clonality markers such as the D816V *KIT* mutation and/or the presence of BM CD25 positive MCs. The proposed diagnosis in these cases is monoclonal mast cell activation syndrome (MCAS) or pre-diagnostic SM [27,33,34].

To avoid false-negative BM assessments, the general recommendation is to address these patients in reference centers with sensitive diagnostic techniques, such as quantitative polymerase chain reaction (PCR)-based molecular assays [3,12,16]. Moreover, multiparametric flow cytometry analysis is also necessary to detect small, atypical MC populations (until 0.001% of CD45 positive BM cells) [10,11].

To exclude SSM or advanced SM without skin lesions, it is important to carefully evaluate the presence of B- and C-findings and the absence of myeloproliferative/myelodysplastic BM features.

In the diagnostic pathway, abnormal sT levels are considered to be a reliable indicator of MC burden, but raised sT levels are not only associated with mastocytosis. However, they might also suggest a hereditary alpha-tryptasemia (HAT) [35,36]. HAT is a biochemical and genetic trait caused by the germline copy number gains of the α-tryptase encoding gene, *TPSAB1*. This genetic alteration was recently reported in about 5% of the healthy population [36]. Although many HAT carriers appear to be asymptomatic, a number of more or fewer specific symptoms are associated with HAT. In particular, symptomatic HAT carriers may refer to similar symptoms to those of MC-mediators of SM, other than a higher risk of anaphylaxis [36]. Therefore, the clinical characteristics of HAT may be close to those of BMM, from which it differs substantially in the absence of clonal BM MCs. It should be noted that HAT has recently been documented in about 17% of patients affected by mastocytosis [37]. Adding the determination of extra copies of *TPSAB1* into diagnostic algorithms for suspected SM patients has been suggested, but its role has not yet been definitively established.

In this review, we aimed at describing the main conditions that should raise the suspicion of mastocytosis, mainly in the absence of typical skin lesions, and address the correct diagnostic strategy.

## 3. Anaphylaxis and BMM

Anaphylaxis is a leading presenting symptom in adult patients with mastocytosis, with a frequency ranging from 20–49% of cases [28,38,39,40]. This range is significantly higher than the 0.05–2% estimated frequency in the general population [41,42]. Furthermore, the frequency of anaphylaxis in patients with BMM is higher than in typical ISM patients with skin lesions. In a study by Alvarez-Twose et al., over 90% of BMM patients had a history of anaphylaxis vs. 24.5% of ISM patients with skin lesions [31]. BMM-associated anaphylaxis was most frequently triggered by a Hymenoptera sting and, less frequently, by ingestion of food or drugs. Moreover, BMM has not infrequently been diagnosed in patients presenting with anaphylaxis without known (idiopathic) triggers [31].

The Spanish Network on Mastocytosis (REMA) found that four clinical elements (male sex, basal sT > 25 ng/mL, and the following characteristics of an allergic reaction: presence of presyncope or syncope and absence of urticaria and angioedema) are independent predictive factors of having a clonal MC disease (CMD) in patients suffering from anaphylaxis or recurrent severe mediator-related symptoms without skin involvement. These parameters, included in the so-called REMA score, are used to identify patients who should undergo BM evaluation because of a high probability of having CMD (Table 2) [31]. Recently, the REMA score has been included in the diagnostic algorithm proposed by the European Competence Network on mastocytosis (ECNM) [16]. The algorithm also included the allele-specific oligonucleotide-qPCR (ASO-qPCR) for D816V *KIT* mutation analysis of blood leucocytes as a screening tool (Figure 1). Unfortunately, the sensitivity of ASO-qPCR for the detection of D816V *KIT* mutation in the peripheral blood (PB) of BMM was significantly lower than that reported in the classical form of ISM with skin lesions (66% vs. 93%) [43]. This data suggests that a combination of the REMA score and sensitive *KIT* D816V mutation analysis of PB is necessary to diagnose underlying BMM in patients with anaphylaxis.

Over the past 15 years, increasing evidence has shown a preferential association between Hymenoptera venom allergy (HVA) and SM:

(a) The prevalence of HVA in SM patients (around 20–30%) is higher than in the general population (0.3–8.9% in the European adult population) [44,45,46].

(b) The Hymenoptera venom sting represents the most common trigger of anaphylaxis in adult mastocytosis patients (22–60% of cases) [28,38,39,40].

(c) The prevalence of mastocytosis in patients with systemic HVA (1–7.9%) is higher than in the general population (1–1.3 cases per 10,000 inhabitants) [47]. The lower prevalence of mastocytosis in patients with HVA reported in some studies could related to the use of low sensitivity screening test [48] or the lack of BM evaluation [49,50,51,52].

When elevated, the basal sT level is a useful criterion for selecting patients with HVA that are eligible for a BM evaluation if SM is suspected [13,53]. Nevertheless, SM cannot be excluded in subjects with severe systemic HVA but normal basal sT [15]. The REMA score exhibits very high sensitivity (91%) and specificity (75%) in screening patients with HVA and suspected mastocytosis without typical skin lesions [31]. The anaphylactic reactions of patients with mastocytosis and HVA are characterized in the majority of cases by the absence of angioedema and erythema and the predominance of cardiovascular symptoms (i.e., hypotension). These symptoms are frequently associated with loss of consciousness [13,15,25,31]. Moreover, most patients do not report MC activation symptoms between acute episodes. Therefore, most of these patients may have HVA severe reactions as the only clinical manifestation of mastocytosis [13,31,38].

Currently, there is no preventive pharmacological treatment available for HVA. Venom immunotherapy (VIT) represents a safe and effective treatment that decreases the risk of subsequent systemic reactions and reduces morbidity and mortality [54]. In the general population, VIT is effective in 77–84% of patients treated with honeybee venom and 91–96% of patients receiving vespid venom [55]. After some debate, regarding safety concerns, it is now generally accepted that VIT is clinically justified in patients with severe HVA and documented mastocytosis [47]. Importantly, these patients should receive life-long VIT. Based on the data currently available, VIT conferred full protection in the majority (86%) of re-stung mastocytosis patients, although this percentage is slightly lower than that reported in patients without SM [54,56].

Interestingly, the probability of a mastocytosis diagnosis is relatively high in patients with HVA that lose VIT protection after a proper course of VIT. Thus, these patients should be investigated for mastocytosis. When a diagnosis of mastocytosis has been established, these patients should continue life-long VIT [47]. Regardless of the sT value, patients with extremely severe reactions to Hymenoptera stings, characterized by hypotension but without urticaria and angioedema, should undergo an accurate hematological workup before terminating immunotherapy.

Patients with mastocytosis and severe systemic reactions should carry two or more epinephrine self-injectors. In the recent European Academy’s position paper, this procedure is also advised for all mastocytosis patients treated with VIT, even if they had reached the maintenance dose [57].

## 4. Osteoporosis and BMM

According to the cohort of Hermans et al., osteoporosis and bone involvement can be the third presenting symptom of MC disorders [58]. Osteoporosis frequently occurs in patients with SM, resulting in fragility fractures with a prevalence of 8–40% for osteoporosis and 3–41% for fractures [30,32,59,60,61,62]. These problems are more evident at the vertebral site either due to low bone mineral density (BMD) or fragility fractures [63]. Thus, the diagnosis of BMM should be considered as a possible cause of secondary bone disease in patients with unexplained fragility fractures, unexplained osteoporosis, or inappropriately low BMD.

In young people or males, inappropriately low BMD might induce the suspicion of a CMD. In these cases, the WHO osteoporosis definition of a T-score < −2.5 can be misleading as it relies on low sensitivity and specificity to detect bone disease. Z-score provides a better evaluation, with a value below −2 suggesting inappropriate BMD and the need for a diagnostic workup to exclude secondary osteoporosis. Similarly, hidden secondary causes should be ruled out in patients with fragility fractures, regardless of the BMD value [63]. In fact, bone involvement in SM is not only a quantitative problem but also a qualitative one, as reflected by fragility fractures that can also occur in the normal and osteopenic range of BMD.

A useful screening tool in these cases is the measurement of sT level (Figure 1). An sT value above 25 ng/mL is strongly suggestive of SM and represents an indication for BM evaluation. Moreover, mildly increased values (15 to 25 ng/mL) may also point to BM biopsy, depending on adjunctive elements of clinical suspicion. Despite the utility of sT as a screening exam, it suffers from false-positive and false-negative cases as does every biomarker. In fact, Carosi et al. reported raised sT levels (>11.4 ng/mL) in 33/232 (14.2%) of a large series of osteoporotic patients, but only 3 out sixteen (19%) patients who agreed to perform the BM evaluation had BMM confirmed [64]. An additional 3 patients (19%) had the D816V *KIT* mutation or CD25 positive MCs, in agreement with a clonal MC disorder. However, these markers are also insufficient for an SM diagnosis. This study displays the need for very sensitive diagnostic tools in this group of patients. Thus, in our opinion, a BM biopsy might also be considered with normal or slightly increased sT if there is a high clinical suspicion, especially in males or patients with a history of anaphylaxis or other typical symptoms (Figure 1).

To note, skin involvement does not seem to affect bone mineral density, but recent evidence suggests that its absence is a risk factor for fractures in SM. Using data from their cohort, a Dutch study group has proposed the “Mast Fx score” as a risk assessment tool for SM patients [60]. The risk factors included in this score are age, male gender, alcohol consumption, Hip T-score (starting from T-score ≤ −1), serum CTX, and lack of skin involvement [60]. The lack of skin involvement as a risk factor probably reflects the higher diagnostic delay and misdiagnosing of this group of BMM.

If osteoporosis and fragility fractures are the most frequent bone manifestations of SM, other signs and symptoms should be mentioned. A minority of patients, especially those with high sT levels or advanced forms of SM [65], usually present with a diffuse osteosclerotic picture which is characterized by high BMD, high bone turnover, and diffuse bone scintigraphy uptake (“superscan”). Sometimes, in the context of a sclerotic bone, small lytic lesions may be identified. A small percentage of patients may also have single or multiple focal sclerotic or lytic lesions [63,66].

## 5. Conclusions

BMM represents an underestimated, indolent SM variant that lacks skin lesions. It is strongly associated with anaphylaxis, mainly related to HVA, or severe osteoporosis as presenting symptom. From a clinical perspective, BMM patients show male prevalence, older age at diagnosis, and fewer mediator-related symptoms (other than anaphylaxis) when compared to typical ISM. Typically, they have slightly elevated or normal sT level and low MC burden and frequency of major histological criterion, requiring a sensitive diagnostic approach in order to avoid false-negative BM assessment. Despite the indolent course, BMM patients are at risk of potentially life-threatening events and severe skeletal complications. Thus, early recognition of BMM is advisable. Moreover, patients with BMM and systemic reactions to Hymenoptera venom should undergo life-long venom immunotherapy.

Given the complexity of the pathology, it is highly recommended to refer these patients to specialized multidisciplinary centers.

## Figures and Tables

**Figure 1 jcm-10-01420-f001:**
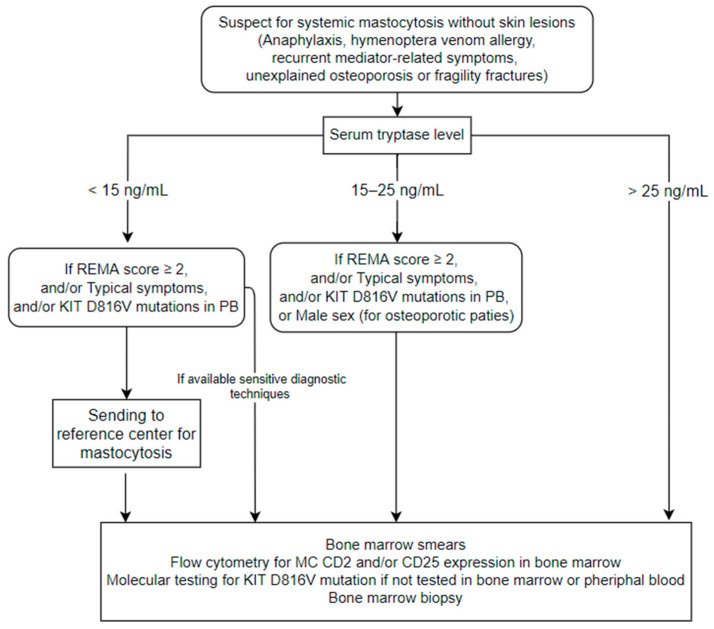
Proposal of diagnostic algorithm in patients with suspicion of Systemic Mastocytosis without typical skin lesions. This figure includes suggestions from both Valent P et al. [16] and Rossini M et al. [32] with some modifications. We used the 3 decision cut-offs of serum tryptase (sT) suggested by Valent et al. (with few differences from those proposed by Rossini et al.) in order to standardize the diagnostic pathway. We included male sex as a criterion only for osteoporotic patients in the cohort with sT between >15 and <25 ng/mL, according to Rossini et al. We also included our personal recommendation of sending patients with sT < 15 ng/mL and with suggested clinical/laboratory characteristics to referral centers if sensitive diagnostic techniques are unavailable. Patients with sT < 25 ng/mL and without criteria should be observed in the follow-up. Abbreviations: REMA: Spanish Network on Mastocytosis, PB: peripheral blood, MC: mast cell.

**Table 1 jcm-10-01420-t001:** World Health Organization (WHO) classification of Systemic Mastocytosis.

	DIAGNOSTIC CRITERIA	B FINDING ^III^	C FINDING ^IV^
	ONE MAJOR ^I^ CRITERIA AND ONE MINOR ^II^ OR THREE MINORS		
BMM	YES	˂2	NO
ISM	YES	˂2	NO
SSM	YES	≥2	NO
ASM	YES		≥1
SM—AHN	YES		ASSOCIATED HEMATOLOGIC NEOPLASM (non MC-lineage)
MCL	≥20% MAST CELL IN BONE MARROW SMEAR		

BMM: bone marrow mastocytosis; ISM: indolent systemic mastocytosis; SSM: smoldering systemic mastocytosis; ASM: aggressive systemic mastocytosis; SM-AHN: systemic mastocytosis with an associated hematological neoplasm; MCL: mast cell leukemia; MC: mast cell; BM: bone marrow. I. Major criteria: histological finding of at least 15 multifocal dense MC infiltrates in BM or other extracutaneous organs. II. Minor criteria: *i.* Abnormal morphology of extracutaneous MCs; *ii.* Serum tryptase >20 ng/mL; *iii.* Expression of CD2 and/or CD25 on MCs in BM or other extracutaneous organs; *iv.* Detection of a mutation at codon 816 of the *KIT* gene in BM, blood or extracutaneous organs. III. B findings: *i.* BM biopsy: >30% infiltration of cellularity by MCs (focal, dense aggregates) and serum total tryptase level >200 ng/mL; *ii.* Signs of dysplasia or myeloproliferation, in non-MC lineage(s), but insufficient criteria for definitive diagnosis of an associated hematological neoplasm (AHN), with normal or only slightly abnormal blood counts; *iii.* Hepatomegaly without impairment of liver function, palpable splenomegaly without hypersplenism, and/or lymphadenopathy on palpation or imaging. IV. C findings: *i.* BM dysfunction caused by neoplastic MC infiltration, manifested by ≥1 cytopenia(s) (ANC < 1.0 × 10^9^/L, Hgb < 10 g/dL, platelet count < 100 × 10^9^/L); *ii*. Palpable hepatomegaly with impairment of liver function, ascites, and/or portal hypertension; *iii*. Skeletal involvement with large osteolytic lesions with/without pathological fractures (pathological fractures caused by osteoporosis do not qualify as a “C” finding); *iv*. Palpable splenomegaly with hypersplenism. *v*. Malabsorption with weight loss due to gastrointestinal mast cell infiltrates.

**Table 2 jcm-10-01420-t002:** REMA score. A score ≥ 2 is an efficient cutoff value to predict both MC clonality and SM.

	VARIABLE	SCORE
GENDER	Male	+1
	Female	−1
CLINICAL SYMPTOMS	Absence of hives, pruritus and angioedema	+1
	Hives, pruritus and angioedema	−2
	Presyncope and/or syncope	+3
S-BASAL TRYPTASE	<15 ng/mL	−1
	>25 ng/mL	+1

Modified from Alvarez-Twose I et al. [25]. Abbreviations: REMA: Spanish Network on Mastocytosis; MC: mast cell; SM: systemic mastocytosis.

## Data Availability

Not applicable.

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
