# Peer review of "Bone Marrow Mastocytosis: A Diagnostic Challenge"

_jcm, 2021, doi:10.3390/jcm10071420_

Round 1

Reviewer 1 Report

This manuscript is informative and is generally written well.

The authors provided detailed information about bone marrow mastocytosis and covered the most recent research articles as well.

Author Response

We acknowledge the Reviewer for his comment. We sincerely hope that our paper could improve the knowledge of this rare disease. 

Reviewer 2 Report

The review by Zanotti and colleagues on Mastocytosis of the Bone Marrow (BMM) is very interesting and useful for doctors working in the field of Mastocytosis. The difficult diagnostic pathway from clinical suspicion to final diagnosis is discussed, underlining the usefulness of referring this kind of patients to specialized and multidisciplinary centers.

I add few comments:

  1. Line 222: "a value above" must be changed to "a sT value above"
  2. Line 148: add a description of different type of elicitors of anaphylaxis in BMM patients.
  3. Add in the paragraph concerning BMM and osteoporosis the data recently reported by Carosi et al (Hypertryptasemia and Mast Cell-Related Disorders in Severe Osteoporotic Patients. Mediators Inflamm. 2020) concerning the incidence of elevated tryptase values ​​in osteoporotic patients and the incidence of BMM diagnosis in the same population.

Author Response

Manuscript ID        jcm-1160377

 Reviewer #2

The review by Zanotti and colleagues on Mastocytosis of the Bone Marrow (BMM) is very interesting and useful for doctors working in the field of Mastocytosis. The difficult diagnostic pathway from clinical suspicion to final diagnosis is discussed, underlining the usefulness of referring this kind of patients to specialized and multidisciplinary centers.

I add few comments:

  1. Line 222: "a value above" must be changed to "a sT value above"
  2. Line 148: add a description of different type of elicitors of anaphylaxis in BMM patients.
  3. Add in the paragraph concerning BMM and osteoporosis the data recently reported by Carosi et al (Hypertryptasemia and Mast Cell-Related Disorders in Severe Osteoporotic Patients. Mediators Inflamm. 2020) concerning the incidence of elevated tryptase values ​​in osteoporotic patients and the incidence of BMM diagnosis in the same population.

 Response to Reviewer #2:

 We thank the Reviewer for these comments, contributing to improve the readability and quality of our manuscript.

  1. We changed this line as the reviewer suggested.
  2. We added a paragraph describing the possible triggers of anaphylaxis (now lines 188-191)
  3. Finally, we have integrated with data about osteoporosis and BMM from the recent study by Carosi et al. (lines 320-325)

Reviewer 3 Report

Dear Authors,

first of all, I thank You for giving me the opportunity to read this Your manuscript. It reviewed the most relevant features and findings of bone marrow mastocytosis (BMM).  

I have only minor comment:

  1. please, remove lines from 34 to 36 because superfluous.
  2. please, remove lines 227-228 because superfluous. The paragraph "Osteoporosis and BMM" should start so : According to the cohorts of Hermans et al.......
  3. In Figure 1, it was not clear what differences there were between the diagnostic algorithm proposed by Rossini et al. and that proposed by Valent et al. Please, clarify this point. 
  4. Please, discuss the relationship between BMM, sarcoma and haematological neoplasms, only mentioned in lines 61-63. In short, how these associations modify clinical findings (if any....), and what warnings can be useful in clinical practice. 
  5. Please, revise spaces in line 174 and in line 230. 
  6. As You correctly highlighted, hereditary alpha-tryptasemia (HAT) is reported in about 5% of the healthy population. Please, discuss differential diagnosis between HAT and BMM. Readers may not be "experts" , and certain concepts should not be taken for granted. What do You think ?      

Author Response

Manuscript ID jcm-1160377

Reviewer #3

First of all, I thank You for giving me the opportunity to read this Your manuscript. It reviewed the most relevant features and findings of bone marrow mastocytosis (BMM).  

I have only minor comment:

  1. please, remove lines from 34 to 36 because superfluous.
  2. please, remove lines 227-228 because superfluous. The paragraph "Osteoporosis and BMM" should start so : According to the cohorts of Hermans et al.......
  3. In Figure 1, it was not clear what differences there were between the diagnostic algorithm proposed by Rossini et al. and that proposed by Valent et al. Please, clarify this point. 
  4. Please, discuss the relationship between BMM, sarcoma and haematological neoplasms, only mentioned in lines 61-63. In short, how these associations modify clinical findings (if any....), and what warnings can be useful in clinical practice. 
  5. Please, revise spaces in line 174 and in line 230. 
  6. As You correctly highlighted, hereditary alpha-tryptasemia (HAT) is reported in about 5% of the healthy population. Please, discuss differential diagnosis between HAT and BMM. Readers may not be "experts", and certain concepts should not be taken for granted. What do You think? 

Response to Reviewer #3:

We thank the Reviewer #3 for the comments and for suggesting us to discuss the differential diagnosis between HAT and BMM. This is a very interesting question that we have tried to address, as you will see in the modified version of the paper.

  1. We modified the text to avoid repetitions.
  2. We removed lines from 227-228 and modified the introduction of the paragraph as suggested.
  3. We have tried to clarify the differences between these two algorithms (lines 259-267) in the legend of the figure
  4. We briefly discussed the differences among BMM, sarcoma and other hematological neoplasms, from a diagnostic point of view (lines 119-124).
  5. We revised spaces.
  6. We discussed this topic in lines 164-175, with more details on the genetics of HAT. We also specify that this condition may present clinical characteristics similar to BMM, from which it differs substantially for the absence of clonal MCs.
